# Detection of nitric oxide-mediated metabolic effects using real-time extracellular flux analysis

Bay Vagher[1,2], Eyal Amiel [1,2]*

1 Cellular, Molecular and Biomedical Sciences Graduate Program, University of Vermont, Burlington, VT, United States of America, 2 The Department of Biomedical and Health Sciences, University of Vermont, Burlington, VT, United States of America

* eyal.amiel@med.uvm.edu

**Data Availability Statement:** All relevant data are within the manuscript and its Supporting Information files.

**Funding:** This work was supported by the National Institutes of Health (NIH), National Institute of

## Abstract

Dendritic cell (DC) activation is marked by key events including: (I) rapid induction and shifting of metabolism favoring glycolysis for generation of biosynthetic metabolic intermediates and (II) large scale changes in gene expression including the upregulation of the antimicrobial enzyme inducible nitric oxide synthase (iNOS) which produces the toxic gas nitric oxide (NO). Historically, acute metabolic reprogramming and NO-mediated effects on cellular metabolism have been studied at specific timepoints during the DC activation process, namely at times before and after NO production. However, no formal method of real time detection of NO-mediated effects on DC metabolism have been fully described. Here, using Real-Time Extracellular Flux Analysis, we experimentally establish the phenomenon of an NO-dependent mitochondrial respiration threshold, which shows how titration of an activating stimulus is inextricably linked to suppression of mitochondrial respiration in an NO-dependent manner. As part of this work, we explore the efficacy of two different iNOS inhibitors in blocking the iNOS reaction kinetically in real time and explore/discuss parameters and considerations for application using Real Time Extracellular Flux Analysis technology. In addition, we show, the temporal relationship between acute metabolic reprogramming and NO-mediated sustained metabolic reprogramming kinetically in single real-time assay. These findings provide a method for detection of NO-mediated metabolic effects in DCs and offer novel insight into the timing of the DC activation process with its associated key metabolic events, revealing a better understanding of the nuances of immune cell biology.

## Introduction

The field of immunometabolism investigates the coordinated and reciprocal regulation of cellular metabolism and immunological outcomes [1,2]. Early inquiries in the field have focused on the phenomenon called "metabolic reprogramming", whereby immune cells fundamentally change their nutrient flux and usage in response to activating signals, thus supporting immune-protective functional outcomes [3–5]. Immune cell activation is also associated with

Allergy and Infectious Diseases P30GM118228 and 1R21AI135385-01A (EA).The funders had no role in study design, data collection and analysis, decision to publish, or preparation of the manuscript.

**Competing interests:** The authors have declared that no competing interests exist.

widespread alterations in gene expression leading to production of inflammatory mediators which aid to sustain metabolic reprogramming throughout the activation process [6–8].

The process of activation and metabolic reprogramming occurs in several different subsets of immune cells such as myeloid cells, which include macrophages and dendritic cells [9,10]. One subset of dendritic cells (DCs) can be modeled using *in vitro*-differentiated bone marrow-derived dendritic cells (BMDCs) and have been shown to induce an acute, rapid, and substantive increase in glucose utilization to support cellular activation [3–5,11], similar to the Warburg effect in cancer contexts [1,12,13]. This shift towards increased glycolytic metabolism marks the first part of metabolic reprogramming, termed acute metabolic reprogramming, during BMDC activation, which is mediated by key transcription factors TBK1/IKKε and results in a significant increase in glycolytic rate, measured by increased hydrogen ion concentrations from the formation of lactate in the cytosol within minutes [3,5].

Along with shifts in metabolism, BMDC activation is associated with major changes in gene expression including the induction of the gene *Nos2*, which encodes for the protein inducible nitric oxide synthase (iNOS) [14–19]. The production of NO is a well-characterized and potent antimicrobial defense mechanism that works by simultaneously biochemically altering the structure and/or function of essential pathogen molecules, for example, rendering [Fe-S] clusters nonfunctional, and impairing pathogen proliferation via blockade of enzymes involved in metabolism [8,20–22]. Although successful as an antimicrobial agent, NO becomes "self-cytotoxic" at high levels and can react with endogenous proteins within the cells that produce it, as well as nearby cells. One category of endogenous NO-mediated self-cytotoxicity is the potent suppression of mitochondrial respiration through reversible inhibition of electron transport chain protein, cytochrome c oxidase (complex IV) and others [23–26].

Historically, characterization of acute metabolic reprogramming and NO-mediated sustained metabolic reprogramming has been documented at fixed timepoints during the activation process, namely immediately after adding stimulus or around 18–24 hours post stimulus, theoretically before and after sufficient levels of NO are in the system to impact cellular metabolism, respectively [3,15,27]. Advances in immunometabolism technologies including the application of Real Time Extracellular Flux Analysis, colloquially referred to as "Seahorse" technology, have increased the specificity and flexibility to interrogate metabolic pathways involved in BMDC activation (Fig 1A and 1B) [4,5] as well as exploring NO-mediated effects on mitochondrial respiration (Fig 1C) [14–16]. The ability to detect and measure real time changes in cellular bioenergetics over a time course and immediately in response to an array of chemical inhibitors and stimulators provides an ideal platform to further investigate BMDC activation-associated metabolic pathways for multi-hour kinetic analysis, which has yet to be documented for NO-mediated mitochondrial respiratory inhibition.

In this following report, we use a dynamic, flexible, and real-time platform to analyze the first 10-hour period during activation whereby we specifically determine when NO-mediated changes in cell metabolism occur, along with showing the versatility of the platform to determine kinetic parameters of various pharmacological inhibitors commonly used in immunometabolism studies of iNOS activity. In addition to immunometabolism-specific studies, these findings provide technical insight for investigators studying mitochondrial respiration, glycolytic dynamics, and secondary processes associated with changes in metabolism in other cell types and systems.

## Materials and methods

### Mice

C57BL/6J mice were all purchase from The Jackson Laboratory and bred at the vivarium at the University of Vermont (Protocol Number: PROTO201900021). Mice were maintained under

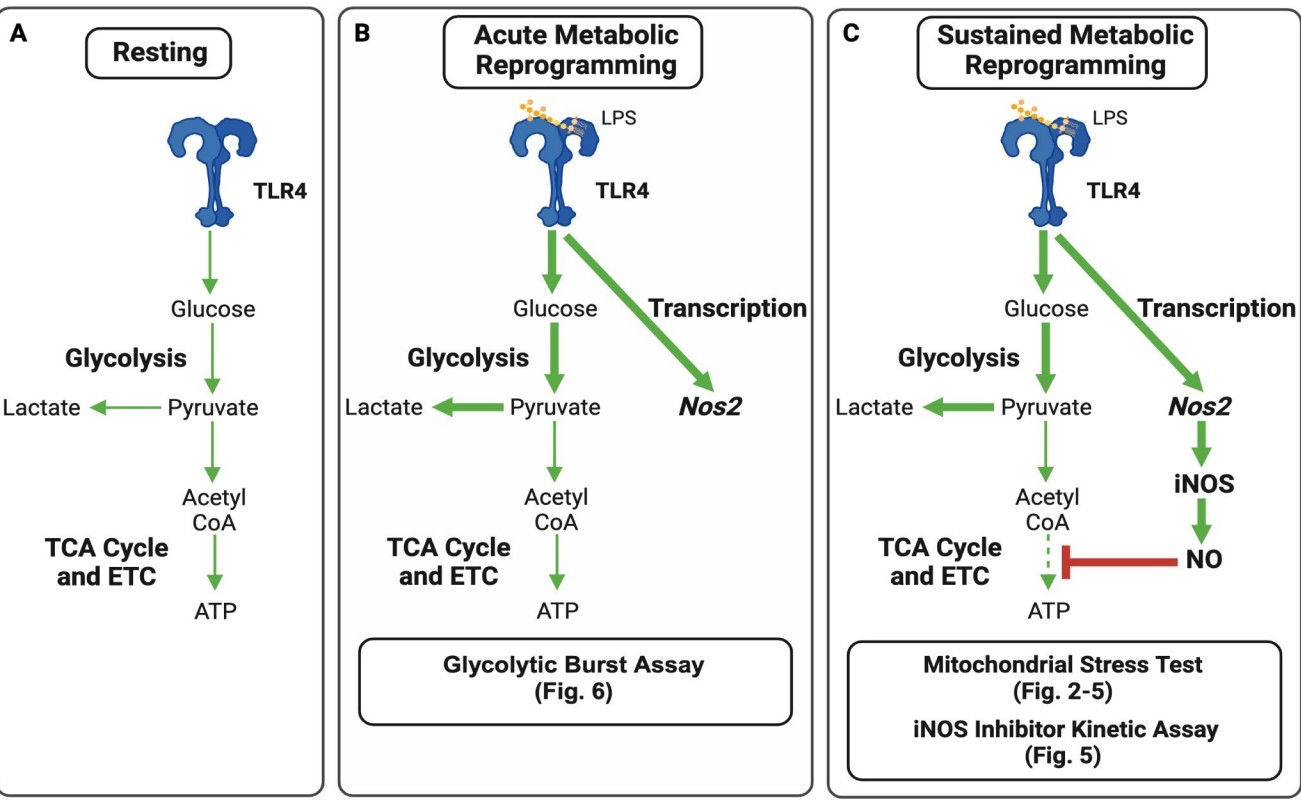

**Fig 1. Simplified overview of key metabolic pathways involved in BMDC activation-associated metabolic reprogramming.** Schematic representations of important metabolic pathways used in (A) resting DCs, (B) during acute metabolic reprogramming events, and (C) during sustained metabolic reprogramming. Assays assessing metabolic changes described in this manuscript are noted where most relevant in the time course of activation.

specific pathogen-free conditions under protocols approved by the Institutional Animal Care and Use Committees. Mice were sacrificed between 6–12 weeks of age using $CO_2$ and cervical dislocation.

## Mouse bone marrow-derived DC culture and activation

Bone marrow-derived DCs were generated as previously described [16]. Briefly, bone marrow cells were differentiated in the presence of recombinant GM-CSF (20 ng/mL, #200–15, Irvine Scientific) in complete DC media (RPMI 1640 containing 10% FBS, 100 U/mL penicillin/ streptomycin, and 2 mM L-glutamine, and 55 mM 2-ME, all from Life Technologies) for 6 d. Cells were seeded at 8 x $10^6$ cells/10 cm plate in 10 mL of complete DC medium. 10 mL of media was added on Day 3 and 15 mL was swapped on Day 5. The TLR4 ligand LPS from *Escherichia coli* 0111:B4 was added at several concentrations indicated in the figures (#tlrl-3pelps, InvivoGen). The iNOS inhibitor S-ethyl-iso-thiourea (SEITU) was used at 500 mM (#81275, Cayman Chemical). The iNOS inhibitor 1400W (HCl) was used at 250 mM (#81520, Cayman Chemical).

## Nitrite quantification

96-well plates were seeded with 2 x $10^5$ cells per well and stimulated for the indicated duration of time. Media supernatant was harvested and run immediately. Nitrite content was measured using the Griess nitrite assay according to manufacturer instructions (#G7921, Molecular

Probes). Quantification of NO indirectly via measurement of the stable intermediate nitrite is a validated and established method as first described here [28–33] as well as macrophage and BMDC models here [14–16,34].

## Real time extracellular flux analysis

**Mitochondrial stress test with iNOS inhibitors.** Seahorse plates were seeded with $2 \times 10^5$ cells per well and stimulated with the indicated LPS doses in ng/mL. Real-time changes in extracellular acidification rates (ECAR) and oxygen consumption rates (OCR) were analyzed using extracellular flux analysis on a Seahorse XFe96 machine (Agilent Cell Analysis, Agilent Seahorse XF Analysis, Santa Clara, CA). In brief, DCs were plated in XF-96 cell culture plates ($2 \times 10^5$ cells/well in 200 μL) and either left unstimulated or stimulated with indicated conditions in complete DC medium. At indicated time points, DCs were switched to and analyzed in XF running buffer (unbuffered RPMI 1640, 5 mM glucose, 10% FBS) per the manufacturer's instructions to obtain real-time measurements of OCR and ECAR. Where indicated, ECAR and/or OCR were analyzed in response to 1 μM oligomycin (Oligo, #J61898MA, Thermo Scientific), 1.5 μM fluoro-carbonyl cyanide phenylhydrazone (FCCP, #15218, Cayman Chemical), and 100 nM rotenone (#150154, MP Biomedicals) plus 1 μM antimycin A (Rot/Ant, #A8674, Sigma Aldrich), 500 μM SEITU, or 250 μM 1400W. All drug dilutions were done in plain XF unbuffered RPMI. The program on Wave went as follows: 3 baseline read cycles, injection 1 (Oligo), 3 read cycles, injection 2 (FCCP), 3 read cycles, injection 3 (Rot/Ant), 3 read cycles, injection 4 (either SEITU or 1400W), 3 read cycles. Each read cycles was 3 minutes of mixing and 3 minutes of measuring for a total time of 2 hours.

**iNOS inhibitor kinetic assay.** Seahorse XF-96 cell culture plates were seeded with $2 \times 10^5$ cells per well concentrated in the center of the plate with two rows of 200 μL cell culture grade water (ccH2O) layering the outside of the plate (4 rows down and 8 columns across, 32 wells total in the middle of the plate) to act as a micro humidifier for evaporation considerations. These considerations are warranted for the Seahorse XFe96 machine and not the Seahorse XF-Pro. Cells were treated with 100 ng/mL LPS for 24 hours before the run. At indicated time points, DCs were analyzed in XF running buffer (unbuffered RPMI 1640, 5 mM glucose, 10% FBS) per the manufacturer's instructions to obtain real-time measurements of OCR and ECAR. Where indicated, ECAR and/or OCR were analyzed in response to 500 μM SEITU, or 250 μM 1400W. All drug dilutions were done in plain XF unbuffered RPMI. The program on Wave went as follows: 6 baseline reads, injection 1 (SEITU or 1400W), 20 read cycles, measurement period 2, 60 read cycles. Each read cycle was 3 minutes of mixing and 3 minutes of measuring for a total time of 8 hours.

**Glycolytic burst assay.** Seahorse XF-96 cell culture plates were seeded with $2 \times 10^5$ cells per well concentrated in the center of the plate with two rows of 200 μL ccH2O layering the outside of the plate (4 rows down and 8 columns across, 32 wells total in the middle of the plate) to act as a micro humidifier for evaporation considerations. These considerations are warranted for the Seahorse XFe96 machine and not the Seahorse XF-Pro. Cells were plated directly after harvest on Day 6 of differentiation. At indicated time points, DCs were analyzed in XF running buffer (unbuffered RPMI 1640, 5 mM glucose, 10% FBS 1 +/- 500 μM SEITU) per the manufacturer's instructions to obtain real-time measurements of OCR and ECAR. Where indicated, ECAR and/or OCR were analyzed in response to 100 ng/mL LPS. All drug dilutions were done in plain XF unbuffered RPMI. The program on Wave went as follows: 6 baseline read cycles, injection 1 (LPS), 20 read cycles, measurement period 2, 92 read cycles. Each read cycle was 3 minutes of mixing and 3 minutes of measuring for a total time of 12 hours. We note here that around 11 hours after the run has started, readings become erratic

due to the chamber being non-humidified, therefore, we chose to only show the first 10 hours of the 12-hour period.

**Quantitative RT-PCR.** 12-well plates were seeded with 2 x $10^6$ cells per well and stimulated for 5 h. Cells were harvested using cell lifters then centrifuged at 500 g for 5 min to pellet. Pellets were stored at -80°C prior to RNA extraction. RNA was extracted according to the manufacturer's instructions using a QIAGEN RNeasy Mini Kit (#74104; QIAGEN). Sample concentration was determined via NanoDrop (NanoDrop 2000 Spectrophotometer; Thermo Fisher Scientific), and 200 ng of RNA was used in the cDNA synthesis reaction according to the manufacturer's instructions using the Verso cDNA Synthesis Kit (#AB1453B; Thermo Scientific). The quantitative RT-PCR (RT-qPCR) reaction was performed according to the manufacturer's instructions for the DyNAmo ColorFlash SYBR Green Kit (#F416S; Thermo Fisher Scientific), using primers specific for *Nos2* (F: AGTTCGTCCCCTTCTCCTGT, R: CCTTGTTC AGCTACGCCTTC), *Tlr4* (F: GGCACTGCATGTGACTTTCC, R: CTCGGCACTTAGCACTGTCA), and *β-actin* (F: AGTGTGACGTTGACATCCGTA, R: GCCAGAGCAGTAATCTCCTTCT). Primers were purchased through Eurofins Genomics. The RT-qPCR reactions were run on a Quant-Studio 3 Real-Time PCR System (96-well, 0.2 ml; #A28137; Thermo Fisher Scientific). We calculated the ΔCt between the housekeeping gene *β-actin* and *Nos2* or *β-actin* and *Tlr4*, then calculated the expression of *Nos2* or *Tlr4* relative to *β-actin* as $2^{(\Delta Ct)}$.

**Flow cytometry.** 96-well plates were seeded with 2 x $10^5$ cells per well and stimulated for the indicated duration of time. Cells were centrifuged at 500 g for 5 minutes. Cell media supernatant was aspirated and saved. Serum-free PBS was added to the wells and cells were put on ice for 10 min prior to transferring to round bottom 96-well plates for staining. Cells were washed with serum-free PBS and resuspended 1:10,000 in fixable Far Red viability stain in PBS for 30 min (#L34973; Molecular Probes/Invitrogen). Cells were washed with Cell Staining Buffer (#420301, BioLegend, hereafter called CSB) at 400 g for 5 min and then fixed in PBS containing 1% formaldehyde for 15 min (#30525-89-4; Alfa Aesar). Single-color controls were prepared in parallel with samples and underwent all incubation and wash steps. For the viability dye single-color control, cells were heat killed at 55°C for 10 min and then pooled in an equal ratio with live cells prior to staining and fixation. Samples were run on an Aurora (Cytek Biosciences). All flow analysis was performed using FlowJo Software version 10.10 (BD Biosciences).

**Western blot.** 12-well plates were seeded with 2 x $10^6$ cells per well and stimulated for 24 hours. Cells were harvested using cell lifters then centrifuged at 500 g for 5 minutes. Pellets were stored at -80°C before use. Once ready to use, pellets were resuspended in Cell Lysis Buffer (#9803; Cell Signaling Technology) containing Pierce Protease and Phosphatase Inhibitors (#A32959; Thermo Fisher Scientific). Samples were kept on ice for 30 min with periodic vortexing. After the time, cells were spun at 9600 g for 10 min, and supernatant was transferred to new tubes. Protein concentration was determined using a Pierce BCA Assay Kit according to manufacturer's instructions (#A32959; Thermo Fisher Scientific), and all samples were normalized to 1 mg/ml in sample buffer containing 2-ME. Samples were heated at 95°C for 5 min prior to loading a 4–15% Mini-PROTEAN TGX Precast Protein Gel (#4561086; Bio-Rad Laboratories) with a molecular weight ladder (#161–0375; Bio-Rad Laboratories). Samples ran at 165V for 30 min, then transferred onto nitrocellulose membranes, according to manufacturer's instructions (#1704270; Bio-Rad Laboratories) using a Trans-Blot Turbo Transfer System (#1704150; Bio-Rad Laboratories). 5% BSA in TBST was used for blocking and all Ab incubations. Overnight incubations in primary Abs iNOS (1:1000; #2982S; Cell Signaling Technology) and β-actin (1:2000; #643802; BioLegend) were followed by washing with TBST and then 1-h incubations in secondary Abs HRP anti-rabbit (1:1000) and HRP anti-mouse (1:5000;

#405306; BioLegend). SuperSignal West Pico PLUS Chemiluminescent Substrate (#4579; Thermo Fisher Scientific) was used for detection, and blots were imaged on a Syngene PXi.

**Quantification and statistical analysis.** Throughout, "n" refers to independent BMDC cultures from individual mice, except for a few replicate experiments in which individual femurs from one mouse were cultured and differentiated separately as biological replicates. Those will be indicated in the figure legends. All statistical tests were performed in GraphPad Prism using t tests or one-way ANOVA on data where appropriate to determine statistical significance. All p-values and tests are indicated on the figures itself. If a p-value is not stated, the data was determined to be non-significant (greater than $p = 0.05$). Pairwise comparisons were added where indicated.

## Results

The role of NO in BMDC activation is a well-established and documented phenomenon that has used extracellular flux technology to measure and analyze suppression of mitochondrial respiration by NO as well as determining the role of both acute and sustained metabolic reprogramming in aiding BMDCs in performing their immune effector functions [14–17]. Despite the rich understanding and use of extracellular flux technology across BMDC immunometabolism studies, formal proof of principle assays to determine the real-time impact of NO-mediated effects by extracellular flux technology have yet to be fully documented and described. Here, we use standard and modified mitochondrial stress tests [15] to pinpoint NO-dependent regulation of the mitochondrial respiration threshold, along with the development and refinement of two assays which aid to characterize the pharmacokinetics of common iNOS inhibitors as well as determine the temporal kinetics of BMDC activation-associated metabolic transitions.

### NO produced via iNOS in BMDCs suppresses mitochondrial respiration in a dose-dependent manner

Foundational studies in BMDC immunometabolism have historically used 100 ng/mL as a standard dose of LPS to stimulate TLR4-specific inflammatory and metabolic outcomes including expression of iNOS, production of NO, and subsequent loss of mitochondrial respiration [3,5,15]. In agreement with these studies, BMDCs stimulated with 100 ng/mL LPS upregulate gene transcription of *Nos2* (Fig 2A), induce iNOS protein expression (Fig 2B), and produce abundant amounts of NO measured indirectly through quantification of nitrite in cell media supernatant (Fig 2C) compared to their unstimulated counterparts. To corroborate previously published findings showing NO-dependent inhibition of mitochondrial respiration [15,16], we activated BMDCs with 100 ng/mL LPS and measured mitochondrial function via a Seahorse mitochondrial stress test. We observed the expected mitochondrial response patterns in unstimulated cells, characterized by increased mitochondrial-linked oxygen consumption (OCR) and consistent with intact and functional mitochondrial function (Fig 2D) [14]. Compared to unstimulated cells, 100 ng/mL LPS stimulated cells exhibited no mitochondrial response pattern to sequential addition of mitochondrial toxins (Fig 2D) as well as significantly reduced mitochondrial-linked OCR showing that the mitochondria are nonfunctional (Fig 2E).

In a related study investigating the genetic regulation of iNOS function, we have recently reported the impact of titrating to lower doses of LPS on iNOS protein abundance, NO production, and subsequent dampening of mitochondrial respiration through an NO-dependent manner [14,16]. Specifically, we describe a discrete level of stimulation which produces sufficient NO to inhibit mitochondrial respiration, which we have called the "mitochondrial respiration threshold", and that subtle modulation of activation signals can regulate whether NO-

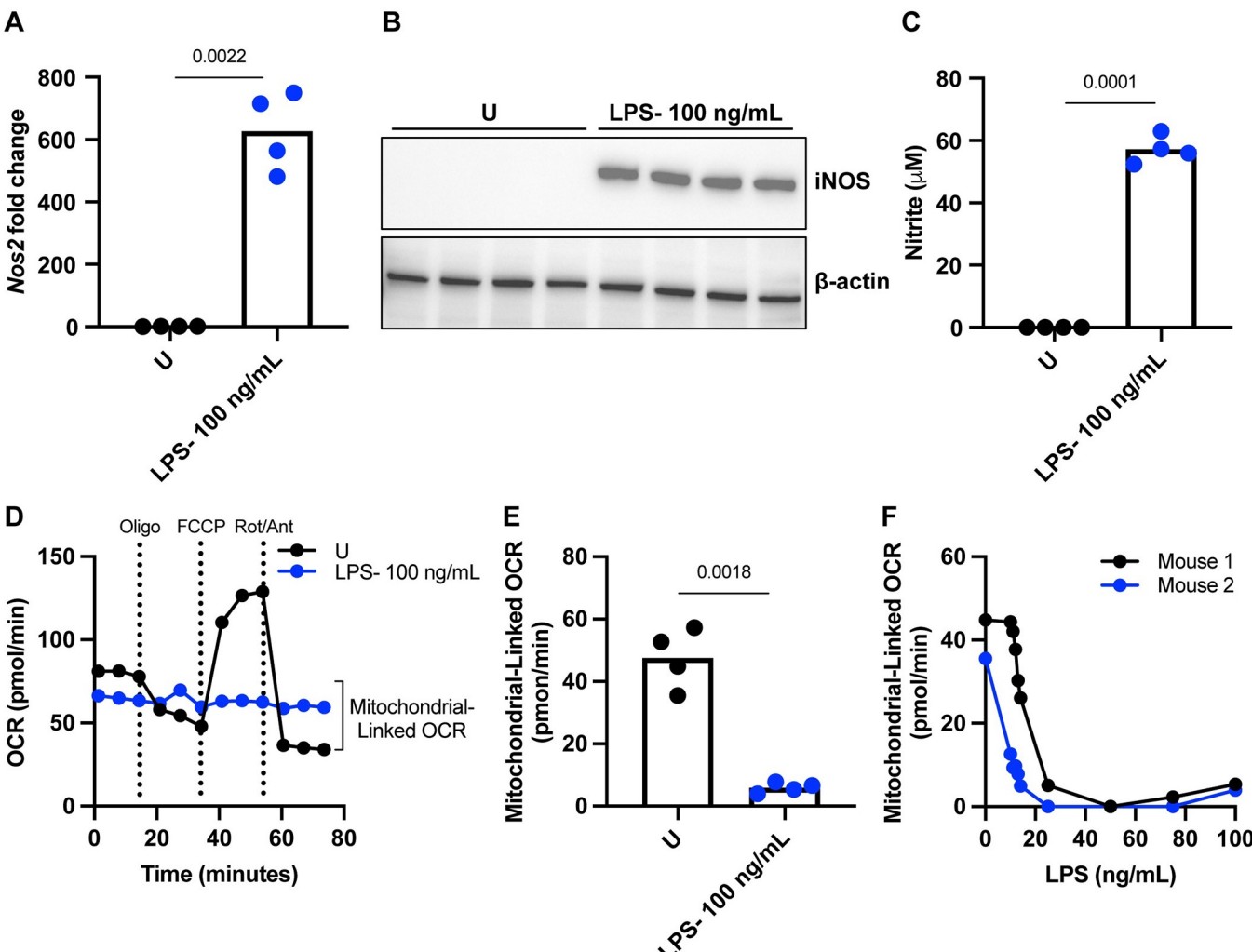

**Fig 2. NO produced via iNOS in BMDCs suppresses mitochondrial respiration in a dose dependent manner.** (A) *Nos2* gene transcription assessed by RT-qPCR from BMDCs stimulated for 5 hours with U or 100 ng/mL LPS. Signals normalized to β-actin as the housekeeping gene via the $2^{(\Delta Ct)}$ method. Analyzed by *t* test, adjusted *p* values are reported (*p* value > 0.05 *ns*), n = 4 biological replicates, representative of at least three independent experiments. (B) Western blot for iNOS from protein lysates isolated from BMDCs stimulated for 24 hours with U or 100 ng/mL LPS. β-actin serves as a loading control, n = 4 biological replicates, representative of at least three independent experiments. (C) Griess nitrite assay on media supernatant from BMDCs stimulated for 24 hours with U or 100 ng/mL LPS. Analyzed by *t* test, adjusted *p* values are reported (*p* value > 0.05 *ns*), n = 4 biological replicates, representative of at least three independent experiments. (D) Mitochondrial stress test OCR kinetic trace of extracellular flux analysis on BMDCs stimulated for 24 hours with U or 100 ng/mL LPS. Oligo, FCCP, and Rot/Ant are short for injections of oligomycin; ATP synthase inhibitor, fluoro-carbonyl cyanide phenylhydrazone; mitochondrial membrane uncoupler, and rotenone/antimycin A; electron transport chain complex I and III inhibitors, respectively. Injections indicated by dashed vertical lines, n = 1 biological replicate, representative of at least three independent experiments. (E) Mitochondrial-linked OCR calculated from kinetic traces in (D) by subtracting the average OCR post inhibition of the electron transport chain by rotenone/antimycin A from average baseline OCR before any injections. Analyzed by *t* test, adjusted *p* values are reported (*p* value > 0.05 *ns*), n = 4 biological replicates, representative of at least three independent experiments. (F) Mitochondrial-linked OCR calculated from kinetic traces during a standard mitochondrial stress test on BMDCs treated with a titration of LPS from 0 ng/mL up to 100 ng/mL, n = 2 biological replicates, representative of three or more independent experiments.

producing cells maintain or lose their respiratory capacity [14,16]. To illustrate the dose-dependent loss of respiration associated with increasing amounts of LPS, we treated BMDCs with a titration of LPS from 0 ng/mL up to 100 ng/mL. We show that as the activating stimulus increased in concentration, there is a discrete level of stimulation that causes a stark drop in mitochondrial respiration at around 25 ng/mL LPS (Fig 2F).

## NO-mediated suppression of mitochondrial respiration occurs at a finite threshold and is directly linked to TLR stimulus concentration

In order to better understand the regulation of the mitochondrial respiration threshold, we set out to investigate the induction of iNOS at the lower doses of LPS. We stimulated BMDCs at lower doses of LPS including 5, 10, and 25 ng/mL. We found that while 5 ng/mL elicited a significant increase in *Nos2* transcription compared to the unstimulated cells, BMDCs stimulated at 25 ng/mL LPS had significantly more *Nos2* transcript than all other doses (Fig 3A). In concordance with *Nos2* transcript levels, while 10 ng/mL LPS showed some iNOS protein, 25 ng/mL LPS had iNOS protein levels in similar density to 100 ng/mL levels (Fig 3B). We also measured induction of Toll-Like Receptor 4 at lower levels of LPS and found that 25 ng/mL LPS and 100 ng/mL LPS cells had two-fold more *Tlr4* transcript than their unstimulated and 5 ng/mL LPS counterparts (S1 Fig).

Fundamental studies investigating the link between NO production and mitochondrial respiration inhibition in BMDCs showed that inhibition of iNOS using iNOS inhibitor SEITU prevents nitrite accumulation in these cells and preserves mitochondrial respiration, thus confirming the direct role of NO production in mitochondrial respiration suppression [15]. To determine if detectable NO accumulation transfers to inhibition of mitochondrial respiration at lower doses of LPS, we again performed a titration of LPS where we measured subsequent production of NO and dampening of mitochondrial respiration via extracellular flux analysis. We found that at lower doses of LPS between 12–14 ng/mL, cells were producing detectable NO around 15–25 μM (Fig 3C). In response to a standard mitochondrial stress test, cells stimulated with 12–14 ng/mL LPS were at intermediate respiration response levels, with a step-wise decrease in mitochondrial respiration as stimulus level increased (Fig 3D). However, at just 10 ng/mL more stimulus, cells treated with 25 ng/mL LPS had significantly higher nitrite accumulation (Fig 3C). The 25 ng/mL BMDCs also exhibited a total loss of mitochondrial responsiveness (Fig 3D) and low mitochondrial-linked OCR (Fig 3E), reminiscent of BMDCs stimulated at 100 ng/mL LPS (Fig 2E).

In congruence with our previous findings [16], we found the amount of NO produced increases correlatively with LPS dose, whereas a drop in oxygen consumption occurs at a specific stimulus "breakpoint" in the LPS dose-titration before leveling out, showing that the relationship between nitrite production and mitochondrial respiration is non-linear. We interpret this to mean that BMDCs producing NO in detectable amounts can have mitochondria that are respiring, however, there is an NO production threshold at which the addition of small amounts of stimulus can increase the amount of NO enough to render these cells functionally non-respiring [16]. This data suggests that modest differences in TLR stimulation levels can influence increases in NO accumulation that put BMDCs over the mitochondrial respiration threshold rendering the mitochondria nonfunctional.

The loss of respiratory capacity via NO inhibitory mechanisms promotes BMDCs towards a metabolic signature characterized by sustained commitment to aerobic glycolysis; however, impaired mitochondrial function eventually results in increased cellular death at 4 days post activation compared to unstimulated control or "sub-threshold-stimulated" BMDCs [5,14]. This activation-associated death is reversible when an iNOS inhibitor was added within 48 hours of activation [14]. Therefore, understanding the nuances of the mitochondrial respiration threshold at low doses of LPS within the first 24–48 hours of activation is predictive of long-term post-activation DC survival [14,16]. To assess long-term post-activation survival in low dose-activated BMDCs, we first measured 96-hour nitrite accumulation via a Griess assay. We show that nitrite levels increased as stimulus amount increased in each biological replicate to a plateau around 25 ng/mL LPS (Fig 3F). To determine if increased NO accumulation conferred to cellular death at 96 hours, we performed flow cytometry. We found that as stimulus

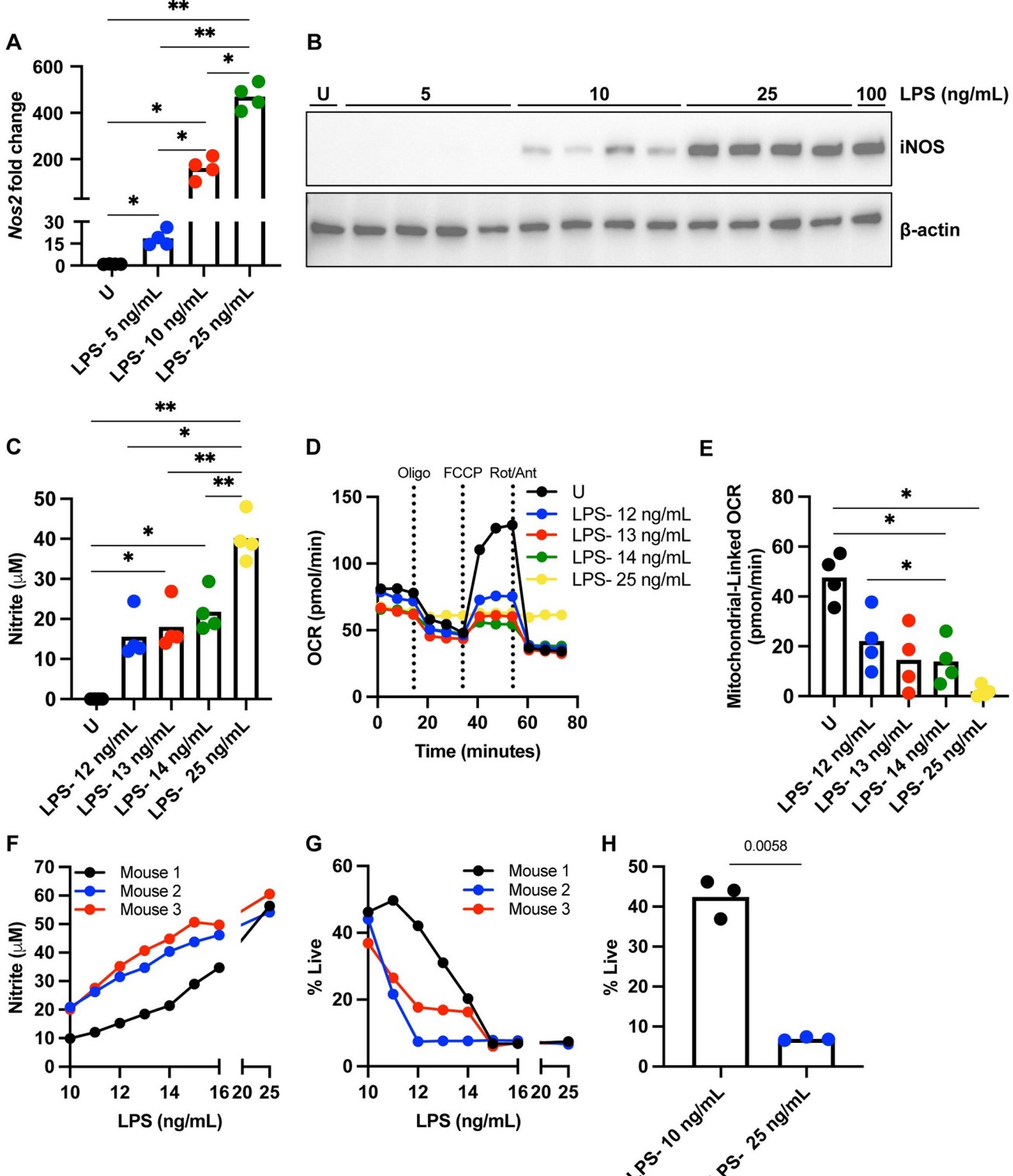

**Fig 3. NO-mediated suppression of mitochondrial respiration occurs at a finite threshold and is directly linked to TLR stimulus concentration.** (A) *Nos2* gene transcription assessed by RT-qPCR from BMDCs stimulated for 5 hours with U, 5 ng/mL, 10 ng/mL, or 25 ng/mL LPS. Signals normalized to β-actin as the housekeeping gene via the $2^{(\Delta Ct)}$ method. Analyzed by one-way ANOVA, adjusted *p* values are reported (*p* value > 0.05 *ns*), n = 4 biological replicates, representative of at least three independent experiments. (B) Western blot for iNOS from protein lysates isolated from BMDCs stimulated for 24 hours with U, 5 ng/mL, 10 ng/mL, or 25 ng/mL LPS. β-actin serves as a loading control, n = 4 biological replicates, representative of at least three

independent experiments. (C) Griess nitrite assay on media supernatant from BMDCs stimulated for 24 hours with U, 12–14 ng/mL, or 25 ng/mL LPS. Analyzed by one-way ANOVA, adjusted $p$ values are reported ($p$ value > 0.05 $ns$), n = 4 biological replicates, representative of at least three independent experiments. (D) Mitochondrial stress test OCR kinetic trace of extracellular flux analysis on BMDCs stimulated for 24 hours with U, 12–14 ng/mL, or 25 ng/mL LPS. Oligo, FCCP, and Rot/Ant are short for injections of oligomycin; ATP synthase inhibitor, fluoro-carbonyl cyanide phenylhydrazone; mitochondrial membrane uncoupler, and rotenone/antimycin A; electron transport chain complex I and III inhibitors, respectively. Injections indicated by dashed vertical lines, n = 1 biological replicate, representative of at least three independent experiments. (E) Mitochondrial-linked OCR calculated from kinetic traces in (D) by subtracting the average OCR post inhibition of the electron transport chain by rotenone/antimycin A from average baseline OCR before any injections. Analyzed by one-way ANOVA, adjusted $p$ values are reported ($p$ value > 0.05 $ns$), n = 4 biological replicates, representative of at least three independent experiments. (F) Griess nitrite assay on media supernatant from BMDCs stimulated for 24 hours with U, 12–14 ng/mL, or 25 ng/mL LPS. Analyzed by one-way ANOVA, adjusted $p$ values are reported ($p$ value > 0.05 $ns$), n = 4 biological replicates, representative of at least three independent experiments. (G) Percentage of live cells assessed by flow cytometry using a fixable Live/Dead stain on BMDCs treated with U, 12–14 ng/mL, or 25 ng/mL LPS for 96 hours, n = 3 biological replicates, representative of at least three independent experiments. (H) Live cell percentage of BMDCs treated with 10 ng/mL and 25 ng/mL LPS for 96 hours from (G). Analyzed by $t$ test, adjusted $p$ values are reported ($p$ value > 0.05 $ns$), n = 3 biological replicates, representative of at least three independent experiments. If no $p$ value is provided, the data is non-significant ($p$ value > 0.05) and therefore, not included on the figure itself. * $p$ < 0.05, ** $p$ < 0.01, *** $p$ < 0.001, **** $p$ < 0.0001.

levels increases, there is a substantial drop in live cell percentage, which eventually plateaus (Fig 3G). When directly comparing lower doses of LPS, a dose of 10 ng/mL elicits production of detectable NO (Fig 3F), however, these cells are still half alive after 96 hours of activation (Fig 3G and 3H). This contrasts with BMDCs stimulated at 25 ng/mL, where only 15 ng/mL LPS produces increased amounts of NO (Fig 3F), and significantly less live cells compared to the 10 ng/mL LPS group (Fig 3H). This data provides a scaffold to begin to understand the regulation of the mitochondrial respiration threshold in more depth.

## Detection of non-mitochondrial oxygen consumption in the iNOS reaction using real time extracellular flux analysis

Extracellular flux analysis technology provides a flexible platform allowing the addition of inhibitors or stimulators as a component of the XF running media and/or as an injection during the assay run. Addition of inhibitors/stimulators in the XF running media provides a way to interrogate baseline real time metabolic differences between control and experimental conditions, and in response to drug injections throughout the course of the assay run. The ability to mix and match how drug inhibitors/stimulators get introduced into the assay provides an effective way to target specific cellular processes of interest and observe the effects of their blockade or promotion on cellular metabolism.

In our model, we can employ iNOS inhibitors in two different ways to interrogate the role of NO in the modulation of cellular metabolism. The first way is when LPS-stimulated BMDCs are treated with SEITU in the XF run media thus blocking the iNOS reaction before the assay is run and throughout the entire assay time course. This is different from adding SEITU as an injection, where LPS-stimulated cells do not lose iNOS functionality until the timed injection at a specified point during the run, allowing visualization of metabolic effects of the inhibitor in real time. The iNOS enzyme catalyzes the reaction where L-arginine, NADPH, and $O_2$ are converted into L-citrulline, NADP$^+$, and NO [18,19]. Therefore, using SEITU as an injection allows interrogation of non-mitochondrial oxygen consumption specific to the iNOS reaction over time. To evaluate the iNOS-linked OCR, we employed a standard Seahorse mitochondrial stress test on both unstimulated and 100 ng/mL BMDCs with the addition of a fourth injection of iNOS inhibitor SEITU. We show that unstimulated cells which respond significantly to electron transport chain complex I and III inhibitors, rotenone and antimycin A, do not respond to SEITU, and have negligible measurable iNOS-linked OCR (Fig 4A and 4B). Whereas 100 ng/mL LPS BMDCs, which are non-responsive to mitochondrial toxins, become inhibited by SEITU within minutes of injection, with their iNOS-linked OCR becoming significantly higher than their unstimulated counterparts (Fig 4A and 4B).

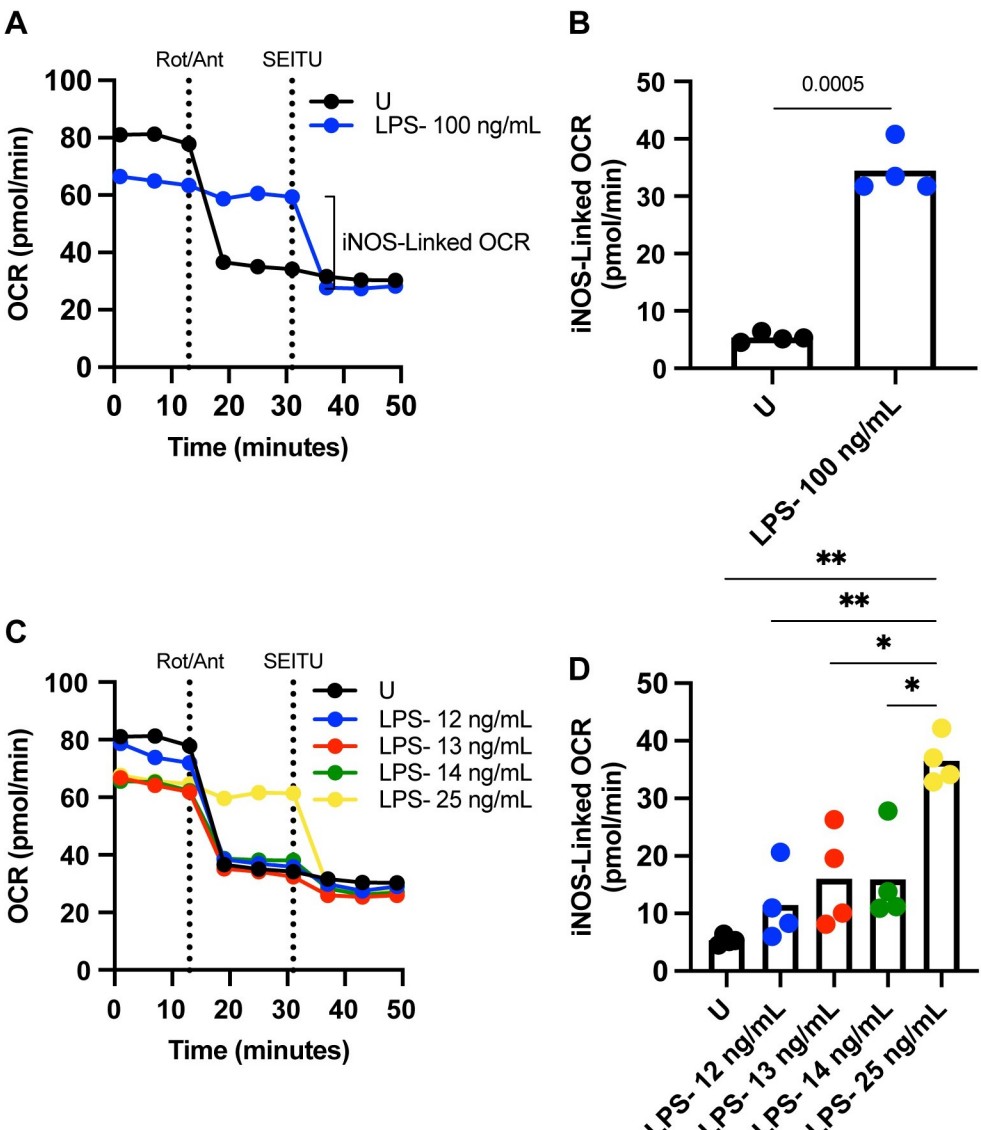

**Fig 4. Detection of non-mitochondrial oxygen consumption in the iNOS reaction using Real Time Extracellular Flux Analysis.** (A) Modified mitochondrial stress test OCR kinetic trace of extracellular flux analysis on BMDCs stimulated for 24 hours with U or 100 ng/mL LPS. Rot/Ant and SEITU are short for injections of Rotenone/Antimycin A; electron transport chain complex I and III inhibitors, and S-ethyl-iso-thiourea; iNOS inhibitor. Injections are indicated by dashed vertical lines, n = 1 biological replicate, representative of at least three independent experiments. (B) iNOS-linked OCR calculated from kinetic traces in (A) by subtracting the average OCR post inhibition of iNOS by SEITU from average baseline OCR before any injections. Analyzed by $t$ test, adjusted $p$ values are reported ($p$ value > 0.05 $ns$), n = 4 biological replicates, representative of at least three independent experiments. (C) Modified mitochondrial stress test OCR kinetic trace of extracellular flux analysis on BMDCs stimulated for 24 hours with U, 12–14 ng/mL, or 25 ng/mL LPS. Injections are indicated by dashed vertical lines, n = 1 biological replicate, representative of at least three independent experiments. (D) iNOS-linked OCR calculated from kinetic traces as in (C) by subtracting the average OCR post inhibition of iNOS by SEITU from average baseline OCR before any injections. Analyzed by one-way ANOVA, adjusted $p$ values are reported ($p$ value > 0.05 $ns$), n = 4 biological replicates, representative of at least three independent experiments. If no $p$ value is provided, the data is non-significant ($p$ value > 0.05) and therefore, not included on the figure itself. * $p < 0.05$, ** $p < 0.01$, *** $p < 0.001$, **** $p < 0.0001$.

At lower doses of LPS, such as 12 ng/mL LPS, BMDCs show similar response patterns and iNOS-linked OCR to unstimulated cells (Fig 4C and 4D). 13 & 14 ng/mL LPS which sit at the cusp of the mitochondrial respiration threshold exhibits slight mitochondrial responsiveness to rotenone/antimycin A compared to unstimulated and 12 ng/mL (Fig 4C), and its iNOS-linked OCR sits notably in the middle of the LPS dose titration (Fig 4D). Comparatively, BMDCs treated with 25 ng/mL LPS are unresponsive to rotenone and antimycin A, but respond to SEITU with a iNOS-linked OCR similar to 100 ng/mL LPS (Fig 4C and 4D). Doses of LPS approaching the respiratory threshold, in this case between 10 ng/mL to 14 ng/mL LPS, can be ideal experimental tools for further studies into molecules or mediators aside from TLR simulation modulation that can push the dose further below or above the threshold. The use of iNOS-linked OCR as a complementary parameter when testing NO-mediated modulation of mitochondrial respiration adds another layer of nuance for interrogation of global metabolic effects of NO in a cellular system.

## Differential kinetics of key iNOS inhibitors SEITU and 1400W

Due to increasing availability and interest in inhibition of NOS enzyme activity, we next tested another commonly used iNOS inhibitor, 1400W, to see if it mirrored our previous results with SEITU. Since SEITU is a reversible pan-NOS enzyme inhibitor, it is an ideal iNOS inhibitor for studying short time points or for experiments where restoration of iNOS function is warranted (Table 1) [35]. 1400W on the other hand is highly specific for the inducible NOS isoform (iNOS) and is irreversible, so that makes it optimal for studying processes where iNOS is the predominant NOS isoform, the enzyme needs to stay completely nonfunctional, or for experiments over extended periods of time (Table 1) [35]. We hypothesized that regardless of the chemical and pharmacological differences between each of the drugs, that they would work similarly in terms of both inhibition of NO production and when used as injections during a mitochondrial stress test. To properly investigate the potency and efficacy of the iNOS inhibitors, we employed the standard historical dose of 100 ng/mL of LPS to ensure maximal induction of iNOS and subsequent NO production.

To test the iNOS inhibitor's ability to shutdown NO production, we stimulated BMDCs in the presence or absence of either iNOS inhibitor then measured their nitrite accumulation 24 hours post activation. LPS induced nitrite accumulation was shutdown to undetectable levels using both iNOS inhibitors (Fig 5A). During BMDC activation, disruption of mitochondrial respiration by NO and its derivatives cause an inability to use the mitochondria to form ATP via ATP synthase [5,14,15]. To investigate whether the iNOS inhibitors restored mitochondrial ATP production in LPS-stimulated BMDCs, we isolated the oligomycin injection on a standard mitochondrial stress test. We report that LPS-stimulated cells in the presence of the iNOS inhibitors responded to oligomycin treatment in a manner close to unstimulated cells (Fig 5B). This carried over to calculation of ATP-linked OCR, where LPS alone-stimulated cells had little to no response to oligomycin where unstimulated, and both iNOS inhibitor treated BMDCs

**Table 1. Binding affinity of common iNOS inhibitors to nitric oxide synthase (NOS) human isoforms.**

| Name | $K_i$ (µM) | | |
|---|---|---|---|
| | nNOS | eNOS | iNOS |
| SEITU | 0.029 | 0.039 | 0.019 |
| 1400W | 2 | 50 | 0.007 ($K_d$) |

Adapted from A.J. Doman et al. 2022 [35].

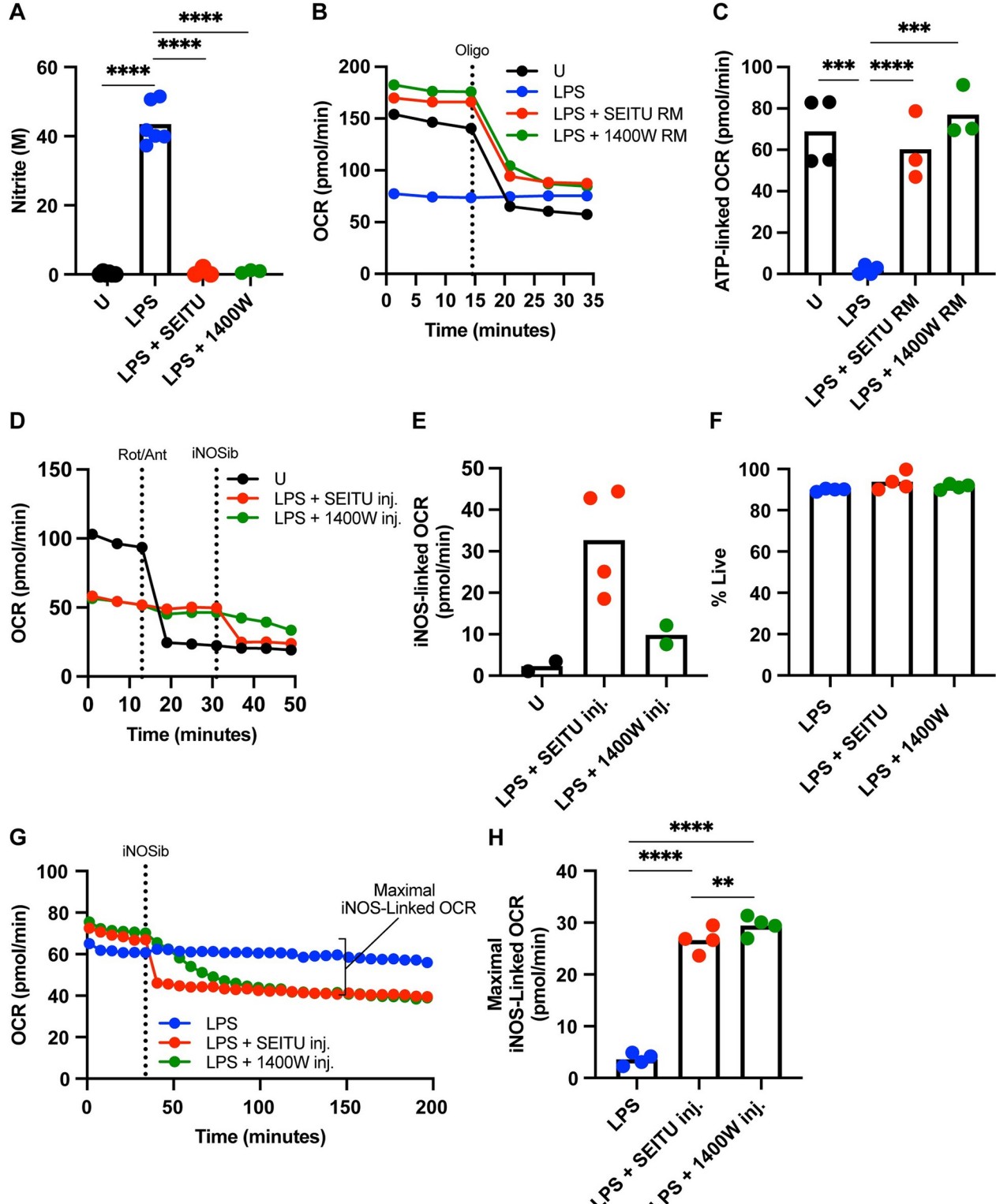

**Fig 5. Differential kinetics of key iNOS inhibitors SEITU and 1400W.** (A) Griess nitrite assay on media supernatant from BMDCs stimulated for 24 hours with U, 100 ng/mL LPS +/- SEITU or 1400W. Analyzed by one-way ANOVA, adjusted *p* values are reported (*p* value > 0.05 *ns*), n = 6 biological replicates for U & LPS-100 ng/mL, and 3 for LPS-100 ng/mL + SEITU or + 1400W groups, representative of at least three independent experiments. (B) Modified mitochondrial stress test OCR kinetic trace of extracellular flux analysis on BMDCs stimulated for 24 hours with U, 100 ng/mL LPS +/- SEITU or 1400W and the iNOS inhibitors in the run media. Oligo is short for injection of oligomycin, ATP synthase inhibitor.

Injection is indicated by dashed vertical lines, n = 1 biological replicate, representative of at least three independent experiments. (C) ATP-linked OCR calculated from kinetic traces in (B) by subtracting the average OCR post injection of oligomycin from average baseline OCR before any injections. Analyzed by one-way ANOVA, adjusted *p* values are reported (*p* value > 0.05 *ns*), n = 4 for U, L100 group, n = 3 for SEITU and 1400W group, representative of three independent experiments. (D) Modified mitochondrial stress test OCR kinetic trace of extracellular flux analysis on BMDCs stimulated for 24 hours with U, 100 ng/mL LPS. Rot/Ant and iNOSib are short for injections of Rotenone/Antimycin A; electron transport chain complex I and III inhibitors, and iNOS inhibitor SEITU or 1400W were indicated in the key. Injections are indicated by dashed vertical lines, n = 1 biological replicate, representative of at least three independent experiments. (E) iNOS-linked OCR calculated from kinetic traces in (D) by subtracting the average OCR post inhibition of iNOS by SEITU or 1400W from average baseline OCR before any injections. n = 4 for SEITU group, n = 2 for 1400W group, representative of three independent experiments. No statistical testing performed due to n = 2 for 1400W group. (F) Percentage of live cells assessed by flow cytometry using a fixable Live/Dead stain on BMDCs treated with 100 ng/mL LPS +/- SEITU or 1400W for 24 hours. Analyzed by one-way ANOVA, adjusted *p* values are reported (*p* value > 0.05 *ns*), n = 4 biological replicates, representative of at least three independent experiments. (G) iNOS inhibitor kinetic assay OCR kinetic trace of extracellular flux analysis on BMDCs stimulated for 24 hours with 100 ng/mL LPS. Injection is indicated by a vertical dashed line, iNOSib is either SEITU or 1400W were indicated in the figure key, n = 1 biological replicate, representative of at least three independent experiments. (H) Maximal iNOS-linked OCR calculated from kinetic traces in (D) by subtracting the average OCR post inhibition of iNOS by SEITU or 1400W at 150 minutes from average baseline OCR before any injections. Analyzed by one-way ANOVA, adjusted *p* values are reported (*p* value > 0.05 *ns*), n = 4 biological replicates, representative of at least three independent experiments. If no *p* value is provided, the data is non-significant (*p* value > 0.05) and therefore, not included on the figure itself. * *p* <0.05, ** *p* <0.01, *** *p* <0.001, **** *p* <0.0001.

had significantly higher ATP-linked OCR values (Fig 5C). Thus, confirming that mitochondrial ATP production is restored when iNOS is inhibited in LPS-stimulated cells.

Since our model postulates that mitochondrial respiration inhibition correlates to discrete levels of NO in the cell, we employed 1400W as a fourth injection on a mitochondrial stress test to calculate the iNOS-linked OCR. Strikingly, compared to the fast drop in oxygen consumption with SEITU, there was little to no response to 1400W (Fig 5D). This lack of response correlated to low levels of iNOS-linked OCR that were comparable to the unstimulated BMDC counterparts (Fig 5E). We did note however, that over the three timepoints measured after the initial 1400W injection, the OCR seemed to be decreasing, however, not to levels comparable to inhibition by SEITU (Fig 5D).

Based on our hypothesis that 1400W would act similarly to SEITU in both the inhibition of NO production and as an injection during a mitochondrial stress test, we speculated that these two inhibitors might exhibit differential inhibitory activity kinetically over time. To ensure that these inhibitors could be used over longer periods of time via extracellular flux analysis, we performed flow cytometry on LPS-stimulated cells in the presence or absence of the two iNOS inhibitors. We show that at 24 hours post-activation, all BMDCs remain mostly alive, confirming no early inhibitor-specific toxicity (Fig 5F).

Previous reporting on the binding speed of 1400W showed that it is indeed a slow binder that requires the cellular cofactor NADPH for complete inhibition [36]. To functionally test and confirm this previous report using extracellular flux technology, we decided to perform what we call an iNOS inhibitor kinetic assay. BMDCs were stimulated with 100 ng/mL for 24 hours and analyzed by extracellular flux analysis. After baseline reads were taken, an injection of SEITU or 1400W was added, then the ensuing drop in OCR was observed for ~4 hours. Interestingly, we showed the initial hard drop in oxygen consumption characteristic of a SEITU injection, and then a slow incremental drop in OCR after the addition of 1400W (Fig 5G). Post injection OCR levels in the SEITU group stayed stable over time, whereas the post injection OCR levels in the 1400W group slowly decreased and leveled out over time reaching the same levels of the SEITU group around the 100-minute mark (Fig 5G).

In the original calculation of iNOS-linked OCR, we took the difference in averages between the three baseline points and the three points immediately after SEITU addition. Due to the differential temporal kinetics of 1400W, we adjusted our iNOS-linked OCR parameter calculations to accommodate the differences in timing. This new parameter called maximal iNOS-linked OCR is calculated by taking the differences in averages between the three baseline

points and the three points flanking the 150-minute mark where OCR is leveled out in both the SEITU and 1400W groups (Fig 5G). With this new parameter the iNOS-linked OCR is better resolved, where it confirms that 1400W is a more potent iNOS inhibitor by the small but significant increase in maximal iNOS-linked OCR compared to the SEITU group (Fig 5H). These results together indicate the need for additional considerations to be taken when choosing what iNOS inhibitor to use for experiments, especially ones that consider timing of inhibition as an important variable.

## Visualization of NO modulation of DC metabolic reprogramming in real time

In response to an array of toll-like receptor (TLR) agonists, BMDCs mount a marked increase in extracellular acidification rate (ECAR) termed the "glycolytic burst," which is required for proper activation and performance of immune effector functions [5]. In DC subsets that do not express iNOS and *in vitro* BDMC cultures where iNOS is genetically deleted, the glycolytic burst occurs and is maintained for some time, until dropping ~4–6 hours post activation [5,14,17]. This suggests that acute glycolytic reprogramming occurs in cells regardless of iNOS expression [5], however, sustained metabolic reprogramming depends on NO-mediated inhibition of many cellular processes including suppression of mitochondrial respiration [15,17]. The two phase model of activation showing acute and sustained metabolic reprogramming have been shown to be related through time, where LPS-stimulated cells mount a glycolytic burst that is sustained, and the addition of an iNOS inhibitor shows where acute metabolic reprogramming ends, and where NO-dependent metabolism changes begin [5]. We set out to build on this model and find a way to pinpoint the specific time when NO begins to exert its effects on metabolic reprogramming and mitochondrial respiration, while also developing parameters that aid one's ability to measure the magnitude of NO respiration inhibition in BMDC activation, all using extracellular flux technology.

Using the basic assay structure as previously published [5], we developed an extended version of the acute glycolytic burst assay. Freshly harvested BMDCs were plated in the middle of a Seahorse cell culture plate. After baseline reads were performed, an injection of LPS 100 ng/mL was administered and the ensuing burst in ECAR was observed over a period of 10 hours; the maximum continuous read-time allotted on the Seahorse XFe96 machine. We then visualized the temporal dynamics of the first 10 hours of BMDC activation (Fig 6A). Akin to the above iNOS inhibitor study, we chose to employ a standard dose of 100 ng/mL LPS to provide a maximal iNOS and NO response. We found that BMDCs activated with LPS, regardless of inhibition of iNOS via iNOS inhibitor SEITU, had a 2–2.5-fold increase in extracellular acidification rate which aligns with previous reports (Fig 6A) [5,15]. This change in glycolytic rate indicates the switch to an emphasis on glycolytic metabolism in formation of lactate using aerobic glycolysis, a hallmark of early BMDC activation [5]. Interestingly, BMDCs stimulated with LPS 100 ng/mL continue to maintain the burst for ~4.5 hours before dropping slightly then picking back up to a level that stays consistent for the rest of the 10-hour period (Fig 6A). The addition of SEITU in LPS 100 ng/mL BMDCs phenocopied responses to LPS 100 ng/mL alone BMDCs during the acute phase, however, ~4.5 hours post activation the ECAR slowly wanes back down towards baseline levels (Fig 6A). To confirm earlier reports that the glycolytic burst occurs irrespective of iNOS and independent of NO modulation [5], we calculated the acute metabolic reprogramming "glycolytic burst" parameter, which shows no difference between groups (Fig 6B), indicating this as an NO-independent phenomenon. In addition, to determine the reliance on NO for maintenance of metabolic reprogramming, we calculated the sustained metabolic reprogramming parameter, which was significantly higher in the LPS

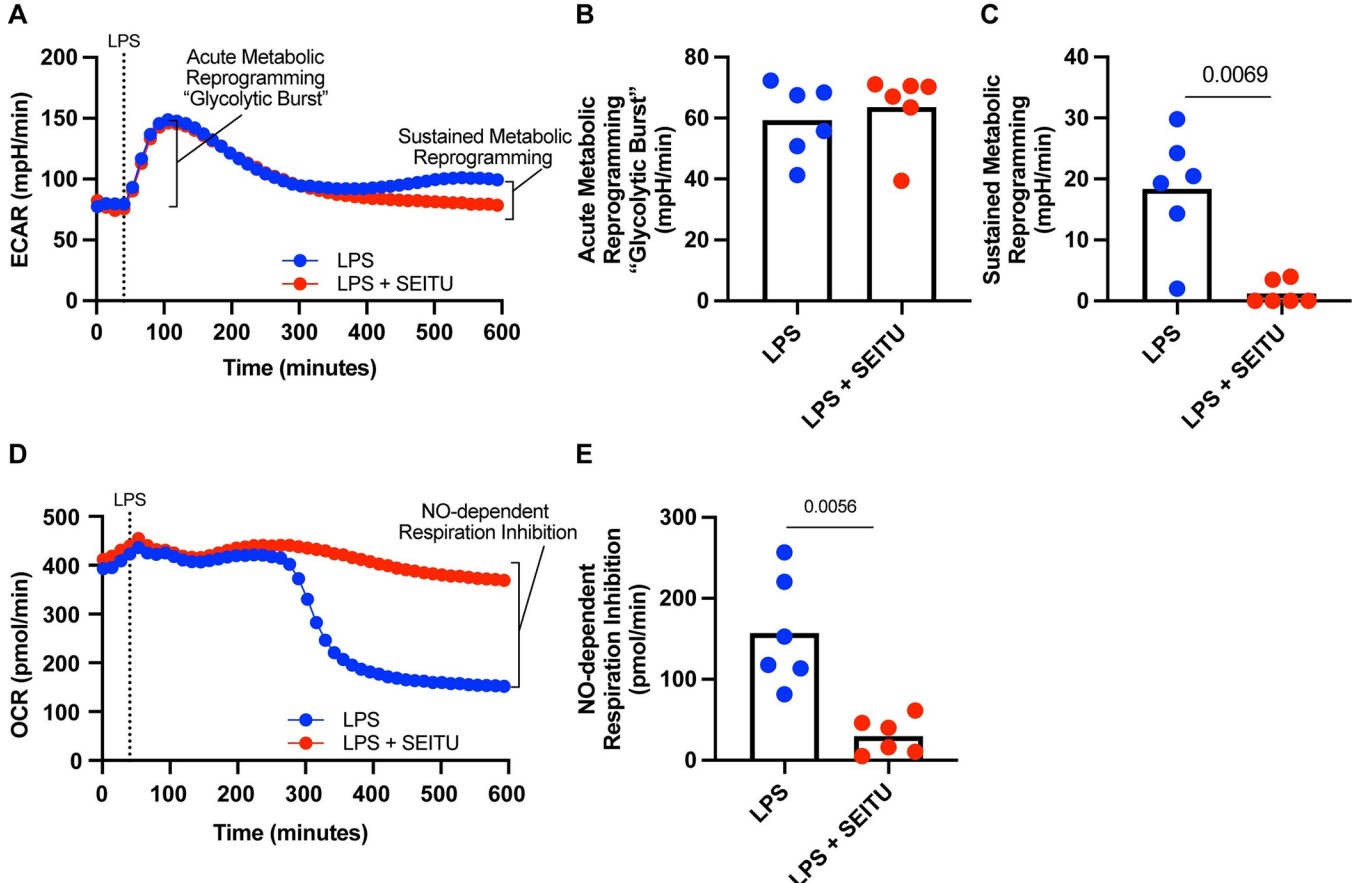

**Fig 6. Visualization of NO modulation of DC metabolic reprogramming in real time.** (A) Glycolytic burst assay ECAR kinetic trace of extracellular flux analysis on freshly harvested BMDCs. Cells were stimulated by injection with 100 ng/mL LPS +/- SEITU. Injection is indicated by dashed a vertical dashed line, n = 1 biological replicate, representative of at least three independent experiments. (B) NO-independent glycolytic burst calculated from kinetic traces in (A) by subtracting the average ECAR post stimulation with LPS at the highest point ~100 minutes from average baseline OCR before any injections. Analyzed by one-way ANOVA using pairwise comparisons by biological replicate, adjusted *p* values are reported (*p* value > 0.05 *ns*), n = 6 independent BMDC cultures from 3 mice, representative of at least three independent experiments. (C) NO-dependent glycolytic reprogramming calculated from kinetic traces in (A) by subtracting the average ECAR post stimulation with LPS at 600 minutes from average baseline OCR before any injections. Analyzed by one-way ANOVA, adjusted *p* values are reported (*p* value > 0.05 *ns*), n = 6 independent BMDC cultures from 3 mice, representative of at least three independent experiments. (D) Glycolytic burst assay OCR kinetic trace of extracellular flux analysis on freshly harvested BMDCs. Cells were stimulated by injection with 100 ng/mL LPS +/- SEITU. Injection is indicated by dashed a vertical dashed line, n = 1 biological replicate, representative of at least three independent experiments. (E) NO-dependent respiration inhibition calculated from kinetic traces in (D) by subtracting the average OCR post stimulation with LPS at 600 minutes from average baseline OCR before any injections. Analyzed by one-way ANOVA using pairwise comparisons by biological replicate, adjusted *p* values are reported (*p* value > 0.05 *ns*), n = 6 independent BMDC cultures from 3 mice, representative of at least three independent experiments. If no *p* value is provided, the data is non-significant (*p* value > 0.05) and therefore, not included on the figure itself.

100 ng/mL alone group (Fig 6C). Therefore, we affirm that while acute metabolic reprogramming is NO-independent, sustained metabolic reprogramming is NO-dependent.

Aside from the role that NO plays in sustaining metabolic reprogramming, NO is a potent inhibitor of mitochondrial respiration. Extracellular flux technology allows for simultaneous measurement of both ECAR and OCR parameters during a single assay run. Thus, during the glycolytic burst assay, we were able to measure metabolic reprogramming through the ECAR parameter, in addition to simultaneous measurement of the temporal kinetics of NO in inhibiting mitochondrial respiration via a decline in the OCR parameter. During the same assay as Fig 6A, OCR measurements were taken for both LPS 100 ng/mL alone and with SEITU added. We show the real time kinetics of mitochondrial respiration suppression in the LPS 100 ng/

mL group where a little before 5 hours there is a substantial decrease in OCR which levels out over time (Fig 6D). Compared to LPS 100 ng/mL alone, the addition of SEITU maintains the OCR over the 10-hour period (Fig 6D). To uphold the role of NO in mediating this decrease in OCR, we calculated the NO-dependent respiration inhibition, where the LPS + SEITU group displays significantly lower amounts of inhibition than the LPS 100 ng/mL alone group (Fig 6E). These data show for the first time the kinetic window during activation where NO accumulates in amounts large enough to inhibit enzymes involved with mitochondrial respiration, which happened earlier than previously documented (Fig 6D) [14,15,17]. Taken together, these data provide a new look into the BMDC activation process including bringing forth new areas of interest into the temporal dynamics of NO production and its inhibition of cellular processes.

## Discussion

DCs can either act as first line defenders or be recruited immune cells to an area of active inflammation [37,38]. Regardless of time of arrival to an area of need, iNOS expressing DCs play an important role in pathogen control through the production of NO [14–17]. The NO-dependent regulation of cellular processes during low level infections or across discrete moments of time when a single microbe interacts with a DC have yet to be experimentally explored. Our work with titration of LPS using mitochondrial stress tests (Fig 7A) represents a more physiologically relevant model, using lower pathogenic molecule stimulus doses. The explicit establishment of a stimulus level-dependent mitochondrial respiration threshold offers new insight into contributing factors to DC metabolic reprogramming which may have major contributions to DC immune effector functions and survival. This work opens opportunities to study what factors and molecules may impact the mitochondrial respiration threshold in a manner that could promote or resolve inflammatory events.

We would also emphasize the importance of performing LPS titrations on each biologically unique BMDC culture before determining the mitochondrial respiration threshold, since our experience suggests that even in cohorts of mice coming from genetically identical backgrounds, the breakpoint of the threshold differs between each biological replicate, across individual cell cultures. However, we consistently see that the patterns of iNOS protein induction, NO production, and dampening of mitochondrial respiration do occur in a consistent

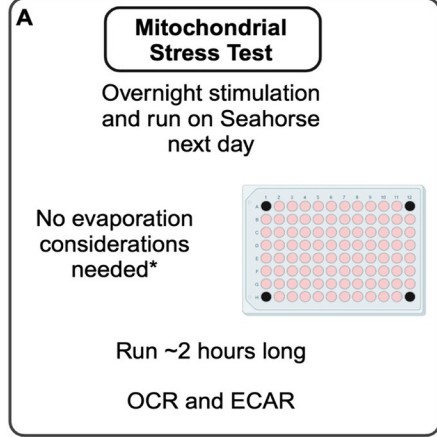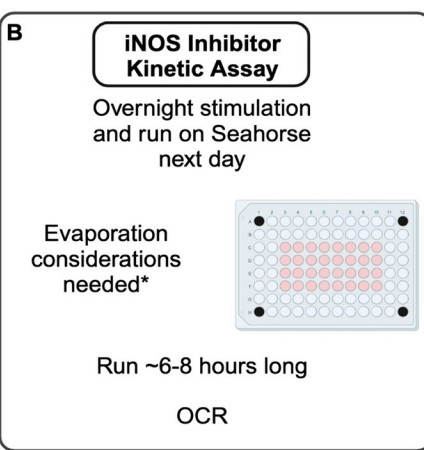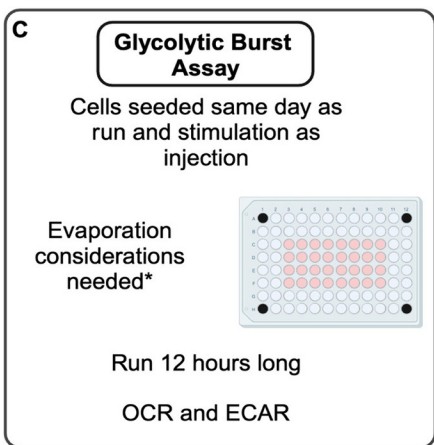

*Evaporation considerations needed for XFe96 model, does not apply to the XF Pro model

**Fig 7. Overview of different assays to assess NO-mediated metabolic effects in DCs.** Schematic representations of experimental and timing considerations for mitochondrial stress tests (A), iNOS inhibitor kinetic assays (B), and glycolytic burst assays (C).

manner, with slightly different LPS doses marking the respiratory threshold across biological replicates [16].

Previous studies using the iNOS-linked OCR parameter showed relatively high amounts of non-mitochondrial iNOS-linked OCR around 75–100 pmol/min. [15], while ours stayed between 20–40 pmol/min. Thus, to best resolve the maximum iNOS-linked OCR, we suggest extending the reads after iNOS inhibitor injection to at least 60–100 minutes to ensure the most accurate picture of the rate of oxygen consumption in the iNOS reaction. Due to new evidence of kinetic differences between iNOS inhibitors, we believe more studies into the sensitivity of other inhibitors used as injections on the Seahorse machine is warranted.

The use of different inhibitors of the iNOS enzyme is ongoing in DC models as well as macrophage models of immunometabolism [25]. In macrophages, studies looking at polarization of macrophage activation states [39,40], determination of NO inhibitory targets during the macrophage activation process [34], and investigation of macrophage pathogen killing capacity to intracellular invaders could be enhanced by our new understanding of iNOS inhibitor kinetics [41,42]. Here, using our iNOS inhibitor kinetic assay (Fig 7B), we confirm 1400W is a slow, and tight binder of iNOS. Interestingly, we postulate that since 1400W requires iNOS reaction cofactor NADPH to reach its full inhibitory effects, that cofactor availability and access could possibly be a reason why 1400W takes longer to work [36].

The early glycolytic burst and switch to using primarily glycolytic metabolism during acute metabolic reprogramming has been shown to be required for proper activation as well as supporting immune effector functions including production of cytokines and initiation of the T cell-mediated immune response [3]. Our lab and others have shown that while glucose is required as a substrate of the burst, the source of the glucose, whether that be extracellular or glycogen-derived, matters [4,5]. We have shown that during the early timepoints of activation, glycogen stores are being broken down in abundant amounts to feed glucose to mounting the glycolytic burst and when you are unable to break glycogen down, the glycolytic rate is not maintained [4]. The glycolytic burst assay would be a helpful tool to be able to delineate further the requirement of glycogen before and after the burst through pharmacological means or via gene-specific deletion of key players in the glycogen metabolic pathway. In addition, it is not known whether loss of the ability to use glycogen-derived carbons impairs the BMDCs ability to mount an NO-dependent sustained metabolic reprogramming regime.

A mechanism by which NO aids to sustain metabolic reprogramming is currently being investigated in macrophages [34]. These authors show direct inhibition of oxidative phosphorylation (OXPHOS) by NO through the coordinated blockade of enzymes involved in the pyruvate dehydrogenase complex, the TCA cycle, and the electron transport chain [34]. Aside from shunting of metabolites towards aerobic glycolysis, NO has been shown to promote accumulation of specific TCA metabolite citrate by targeted inhibition of aconitase 2 in macrophages [34]. While the reason for accumulation of citrate has yet to be fully understood [34,43,44], we do believe that the inhibition of mitochondrial respiration by NO remains geared towards long term survival of LPS-stimulated BMDCs rather than to generate biosynthetic intermediates [14,15]. Stimulated BMDCs continuously produce NO, however, sustained inhibition of the mitochondria eventually results in NO-dependent cell death [14,16]. It is unclear the mechanism by which continuous suppression of OXPHOS in BMDCs is maintained. We believe it could be multifactorial and dependent on substrate availability, the time window during activation, as well as the possible presence of a positive feedback loop for NO where NO itself promotes its own renewal. Determination of how this NO self-renewal positive feedback loop works such as at the transcriptional, translational, or functional level are still ongoing. Regardless of the mechanism, the determination of the mitochondrial respiration threshold as well as the temporal dynamics involved in suppression of mitochondrial

respiration via the glycolytic burst assay could provide useful means to investigate these questions further.

Our work suggests that acute and sustained metabolic reprogramming are linked throughout time under a continuum of post-activation metabolic reprogramming. We show the two phases of metabolic reprogramming in a single kinetic assay and identify the timing in which NO begins to elicit its effects on both glycolytic reprogramming and inhibition of mitochondrial respiration using our glycolytic burst assay (Fig 7C). The mechanism behind the events preceding mitochondrial respiration inhibition and shunting of glucose-derived carbons towards lactate production are still ongoing. Our work uncovers novel timing considerations that may influence future studies into these processes. Additional research into which event, NO-related effects on glycolytic reprogramming or dampening of mitochondrial respiration, happens first in the chain of events would help to further define the mechanism. We believe these assays can be applied to the discovery of other upstream modulators of iNOS aside from TLR signaling who may work to speed up NO-related effects on metabolism and/or increase their amplitude.

Overall, our work seeks to delineate a new layer of interrogation of NO-mediated effects on BMDC immunometabolism using Real-Time Extracellular Flux Analysis. We have shown here the versatility and flexibility of the platform in determining the mitochondrial respiration threshold, non-mitochondrial oxygen consumption in the iNOS reaction, kinetic considerations of common iNOS inhibitors, and helping us visualize the important beginnings of metabolic reprogramming and mitochondrial respiration inhibition by NO during BMDC activation. We believe this work extends outside of immune cell biology, to provide other means of assessment of metabolic reprogramming and NO-related modulation of cellular processes, which could be applicable to other cell types and tissues of scientific interest.

## Supporting information

**S1 Fig. Transcription of gene for upstream signaling modulator TLR4 at lower levels of TLR stimulation.** (A) *Tlr4* gene transcription assessed by RT-qPCR from BMDCs stimulated for 5 hours with U, 5 ng/mL, 10 ng/mL, or 25 ng/mL LPS. Signals normalized to β-actin as the housekeeping gene via the $2^{(\Delta Ct)}$ method. Analyzed by one-way ANOVA, adjusted *p* values are reported (*p* value $> 0.05$ *ns*), n = 4 biological replicates, representative of at least three independent experiments. * $p < 0.05$, ** $p < 0.01$, *** $p < 0.001$, **** $p < 0.0001$.
(TIF)

**S1 Protocol. Extended materials and methods for all extracellular flux analysis assays: Modified mitochondrial stress test, iNOS inhibitor kinetic assay, and glycolytic burst assay.**
(PDF)

**S1 Raw images.**
(TIF)

## Acknowledgments

Special thank you and appreciation to Dr. Ralph Budd and Dr. Dimitry Krementsov from the Vermont Center for Immunology and Infectious Disease, Dr. Matt Poynter from the University of Vermont Department of Medicine, and the whole University of Vermont Biomedical and Health Sciences Department for your intellectual theorizing, experimental assistance, and life advice. Additional love to my dude Jeff from Agilent, you rock!

## Author Contributions

**Conceptualization:** Bay Vagher, Eyal Amiel.

**Data curation:** Bay Vagher.

**Formal analysis:** Bay Vagher.

**Funding acquisition:** Eyal Amiel.

**Investigation:** Bay Vagher.

**Methodology:** Bay Vagher, Eyal Amiel.

**Supervision:** Eyal Amiel.

**Writing – original draft:** Bay Vagher.

**Writing – review & editing:** Bay Vagher, Eyal Amiel.

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
