## [Decision Letter · Decision Letter 0]

25 Sep 2023

PONE-D-23-29008Detection of nitric oxide-mediated metabolic effects using real-time extracellular flux analysisPLOS ONE

Dear Dr. Amiel,

Thank you for submitting your manuscript to PLOS ONE. After careful consideration, we feel that it has merit but does not fully meet PLOS ONE’s publication criteria as it currently stands. Therefore, we invite you to submit a revised version of the manuscript that addresses the points raised during the review process.

We look forward to receiving your revised manuscript.

Kind regards,

Nisha Singh, Ph.D.

Academic Editor

PLOS ONE

Journal Requirements:

4. Thank you for stating the following financial disclosure: "This work was supported by the National Institutes of Health (NIH), National Institute of Allergy and Infectious Diseases P30GM118228 and 1R21AI135385-01A (EA)".

Reviewers' comments:

Reviewer's Responses to Questions

**Comments to the Author**

1. Is the manuscript technically sound, and do the data support the conclusions?

Reviewer #1: Yes

Reviewer #2: Yes

2. Has the statistical analysis been performed appropriately and rigorously? 

Reviewer #1: Yes

Reviewer #2: Yes

3. Have the authors made all data underlying the findings in their manuscript fully available?

Reviewer #1: Yes

Reviewer #2: No

4. Is the manuscript presented in an intelligible fashion and written in standard English?

Reviewer #1: Yes

Reviewer #2: Yes

5. Review Comments to the Author

Reviewer #1: Seahorse technology has been a common and versatile tool to study glycolysis and oxidative phosphorylation in a broad range of disease settings. While this method does contribute to the acute changes in the energy demand the changes associated with cell differentiation are likely to be different. This excellent paper addresses real-time platforms to analyze metabolic changes involved in BMDC activation with LPS endotoxin for the first 10-hour period under the influence of NO-mediated changes. It is impressive that the Eyal group used reversible and irreversible NO inhibitors to study iNOS activity which expand the knowledge of inhibitors choice to the field of interest. Overall, this manuscript is well-written, and it provides a valuable contribution to the field. However, as this article may likely serve as a blueprint/guide for future publications, a few points need to be addressed to make it even more comprehensive and usable for a broad audience.

Figure1. Authors claimed mitochondrial respiration suppression occurs at a finite LPS concentration and is directly linked with TLR stimulation. I have some concerns about these conclusions:

1. LPS activates BMDC which is well-established in the field, and it has been shown by the same group in the previous studies as well. It has also been included that a dose of LPS approaching to respiratory threshold can be an ideal experimental concentration, but which concentration is missing? However, why did the authors choose 100 ng/ml concentration in Fig 1A-C- as well as in the subsequent figures, the rationale is missing (LPS at 100 ng/ml completely blocks mitochondrial respiration meaning around 45 uM Nitrite concentration is sufficient, but in Fig 1D-E even very low concentration like 3ng/mL (20 uM nitrite concentration) is also sufficient to suppress mitochondrial function- jumping from 100 to a range of 1 to 5 ng/mL is confusing). This figure concludes the inhibitory role of LPS on mitochondrial function at different doses of LPS only. To address this author can use multiple concentrations of LPS and measure NO level along with a cell death assay from the same treatment group to confirm that the used concentration is physiologically relevant and sufficient at a finite LPS concentration.

2. iNOS western blot in the LPS-treated cells at different concentrations of LPS will be informative, but not necessary. It will correlate the transcription and translation in a time-dependent manner.

3. Why 2ng/mL LPS OCR is high compared to U whereas other concentrations show inhibitory activity.

4. LPS is a well-known ligand for TLR4 but how different doses of LPS influence the expression of TLR is missing.

Figure 2. Authors claimed detection of NO-mediated modulation of mitochondrial respiration in real time suggests its global metabolic effects. I have some suggestions for clarity:

1. Line 273 to 275 and line 289 to 291 is confusing. SEITU was added in XF run media or was SEITU used as an injection? How the author’s model differs from the old model requires clarity. Please simplify for better understanding to the readers.

2. Fig 2C- why does 3 ng/mL show mitochondrial responsiveness not like other concentrations?

Figure 3. Authors compared reversible vs irreversible iNOS inhibitors which could be important to the field of infection biology as well as could be an indicator of NADPH level for further experimentation.

1. Here author should add no difference in cell death with and without both inhibitors.

2. How is ATP production affected using both inhibitors and can measured in real time as well? This is easy to achieve.

Figure 4. In this figure authors compared ECAR vs OCR with LPS and in the presence or absence of SEITU. They observed metabolic reprogramming in BMDC is NO-independent while sustained reprogramming is NO-dependent.

1. Fig 4B is not informative since Fig 4A itself explains no change in a glycolytic burst between the two groups (this can go to the supplementary figure). While Fig 2C is quantitatively good for understanding the exact difference between the two groups.

2. Of curiosity did authors go longer than 600 minutes for ECAR and OCR and find any difference?

Reviewer #2: In the manuscript “Detection of nitric oxide-mediated metabolic effects using real-time extracellular flux

Analysis” the authors established the phenomenon of an NO-dependent mitochondrial respiration threshold, which is linked to suppression of mitochondrial respiration in an NO-dependent manner. Authors further explored the efficacy of two different iNOS inhibitors in blocking the iNOS reaction kinetically in real time and explored parameters for application using Real Time Extracellular Flux Analysis. The experiments are conducted well but authors need to clarify a few points to strengthen the outcomes of the paper.

1) Is nitrite quantification an established method? Please provide few citations in the method section.

2) Glycolytic burst is measured only based on pH. Did author consider measuring the concentration of extracellular and intracellular lactate? This will provide direct glycolytic flux because of the production of lactate. Kindly measure the lactate and Alanine concentration for the groups.

3) Make a schematic diagram of the overall research plan for this study. That will clearly demonstrate the various experimental events.

4) Did author measure the viability of cell at the end of OCAR and ECAR assays? Since these assays are really long, there are some chances that cells might chew up all of the essential components of media and become nutrition-deficient and might start slowing down metabolic activities and dying.

6. PLOS authors have the option to publish the peer review history of their article (what does this mean?). If published, this will include your full peer review and any attached files.

Reviewer #1: No

Reviewer #2: No

---

## [Author Response · Author response to Decision Letter 0]

9 Jan 2024

January 7th, 2024

Dear Editors and Reviewers of PLOS One,

Please find below a point-by-point response (in blue font) to each reviewer’s critique (italicized) outlined below:

Editors Comments:

3. We note that the grant information you provided in the ‘Funding Information’ and ‘Financial Disclosure’ sections do not match. When you resubmit, please ensure that you provide the correct grant numbers for the awards you received for your study in the ‘Funding Information’ section. I need your help to edit these.

The Funding information and Financial Disclosure sections should now match in the resubmission information.

4. Thank you for stating the following financial disclosure: "This work was supported by the National Institutes of Health (NIH), National Institute of Allergy and Infectious Diseases P30GM118228 and 1R21AI135385-01A (EA)".

Please include this amended Role of Funder statement in your cover letter; we will change the online submission form on your behalf. I need your help with this.

Reviewers' comments:

Reviewer's Responses to Questions

Comments to the Author

3. Have the authors made all data underlying the findings in their manuscript fully available?

Reviewer #1: Yes

Reviewer #2: No

We are not clear on the basis of Reviewer #2’s response to this question and have endeavored to fulfill all of the Data Policy requirement for the PLOS submission process.

5. Review Comments to the Author

Reviewer #1: Seahorse technology has been a common and versatile tool to study glycolysis and oxidative phosphorylation in a broad range of disease settings. While this method does contribute to the acute changes in the energy demand the changes associated with cell differentiation are likely to be different. This excellent paper addresses real-time platforms to analyze metabolic changes involved in BMDC activation with LPS endotoxin for the first 10-hour period under the influence of NO-mediated changes. It is impressive that the Eyal group used reversible and irreversible NO inhibitors to study iNOS activity which expand the knowledge of inhibitors choice to the field of interest. Overall, this manuscript is well-written, and it provides a valuable contribution to the field. However, as this article may likely serve as a blueprint/guide for future publications, a few points need to be addressed to make it even more comprehensive and usable for a broad audience.

We are grateful for Reviewer #1’s favorable assessment of the quality and significance of the findings in this manuscript.

Figure1. Authors claimed mitochondrial respiration suppression occurs at a finite LPS concentration and is directly linked with TLR stimulation. I have some concerns about these conclusions:

1. LPS activates BMDC which is well-established in the field, and it has been shown by the same group in the previous studies as well. It has also been included that a dose of LPS approaching to respiratory threshold can be an ideal experimental concentration, but which concentration is missing? This information is now included on line 436. However, why did the authors choose 100 ng/ml concentration in Fig 1A-C (this is now Figure 2) as well as in the subsequent figures, the rationale is missing (LPS at 100 ng/ml completely blocks mitochondrial respiration meaning around 45 uM Nitrite concentration is sufficient, but in Fig 1D-E (this now Figure 3) even very low concentration like 3ng/mL (20 uM nitrite concentration) is also sufficient to suppress mitochondrial function- jumping from 100 to a range of 1 to 5 ng/mL is confusing). The rationale for beginning these studies at the 100 ng/mL LPS dose is now provided on lines 245-248, in reference to Figure 2: “Foundational studies in BMDC immunometabolism have historically used 100 ng/mL as a standard dose of LPS to stimulate TLR4-specific inflammatory and metabolic outcomes including expression of iNOS, production of NO, and subsequent loss of mitochondrial respiration”. In reference to Figure 5 on lines 453-455, we add: “To properly investigate the potency and efficacy of the iNOS inhibitors, we employed the standard historical dose of 100 ng/mL of LPS to ensure maximal induction of iNOS and subsequent NO production.”. In reference to Figure 7 on lines 569-571, we write: “Akin to the above iNOS inhibitor study, we chose to employ a standard dose of 100 ng/mL LPS to provide a maximal iNOS and NO response.”.

The rationale for titrating lower doses of LPS is now provided on lines 286-293 & 297-298, where we write: “Specifically, we describe a discrete level of stimulation which produces sufficient NO to inhibit mitochondrial respiration, which we have called the “mitochondrial respiration threshold”, and that subtle modulation of activation signals can regulate whether NO-producing cells maintain or lose their respiratory capacity. To illustrate the dose-dependent loss of respiration associated with increasing amounts of LPS, we treated BMDCs with a titration of LPS from 0 ng/mL up to 100 ng/mL. We show that as the activating stimulus increased in concentration, there is a discrete level of stimulation that causes a stark drop in mitochondrial respiration at around 25 ng/mL LPS (Fig 2F).” and “In order to better understand the regulation of the mitochondrial respiration threshold, we set out to investigate the induction of iNOS at the lower doses of LPS.” 

This figure concludes the inhibitory role of LPS on mitochondrial function at different doses of LPS only. To address this author can use multiple concentrations of LPS and measure NO level along with a cell death assay from the same treatment group to confirm that the used concentration is physiologically relevant and sufficient at a finite LPS concentration. These requested data are now provided in the new figure 3F and 3G.

2. iNOS western blot in the LPS-treated cells at different concentrations of LPS will be informative, but not necessary. It will correlate the transcription and translation in a time-dependent manner. The requested data are now included in Figure 2A & B and Figure 3A & B, and describe in the manuscript in lines 248-249 and 298-303.

3. Why 2ng/mL LPS OCR is high compared to U whereas other concentrations show inhibitory activity. This concern is no longer applicable to the new figures provided for this resubmission.

4. LPS is a well-known ligand for TLR4 but how different doses of LPS influence the expression of TLR is missing. We now provide the requested TLR4 gene expression data in Supplemental Figure 1A, described in lines 303-306 in the manuscript text.

Figure 2. Authors claimed detection of NO-mediated modulation of mitochondrial respiration in real time suggests its global metabolic effects. I have some suggestions for clarity:

1. Line 273 to 275 and line 289 to 291 is confusing. SEITU was added in XF run media or was SEITU used as an injection? How the author’s model differs from the old model requires clarity. Please simplify for better understanding to the readers. The relevant sections are now on lines 393-398, where we provide this clarification: “In our model, we employ iNOS inhibitors in two different ways to interrogate the role of NO in the modulation of cellular metabolism. First, LPS-stimulated BMDCs are treated with SEITU in the XF run media thus blocking the iNOS reaction before the assay is run and throughout the entire assay time course. This is different from adding SEITU as an injection, where LPS-stimulated cells do not lose iNOS functionality until the timed injection at a specified point during the run, allowing visualization of metabolic effects of the inhibitor in real time.”

2. Fig 2C- why does 3 ng/mL show mitochondrial responsiveness not like other concentrations? This concern is no longer applicable to the new figures provided for this resubmission.

Figure 3. Authors compared reversible vs irreversible iNOS inhibitors which could be important to the field of infection biology as well as could be an indicator of NADPH level for further experimentation.

1. Here author should add no difference in cell death with and without both inhibitors. The requested data are now included as Figure 5F and described on lines 518-522: “To ensure that these inhibitors could be used over longer periods of time via extracellular flux analysis, we performed flow cytometry on LPS-stimulated cells in the presence or absence of the two iNOS inhibitors. We show that at 24 hours post-activation, all BMDCs remain mostly alive, confirming no early inhibitor-specific toxicity (Fig 5F).”

2. How is ATP production affected using both inhibitors and can measured in real time as well? This is easy to achieve. The requested data are now included as Figure 5B & C and described in lines 463-472 of the manuscript: “During BMDC activation, disruption of mitochondrial respiration by NO and its derivatives cause an inability to use the mitochondria to form ATP via ATP synthase. To investigate whether the iNOS inhibitors restored mitochondrial ATP production in LPS-stimulated BMDCs, we isolated the oligomycin injection on a standard mitochondrial stress test. We report that LPS-stimulated cells in the presence of the iNOS inhibitors responded to oligomycin treatment in a manner close to unstimulated cells (Fig 5B). This carried over to calculation of ATP-linked OCR, where LPS alone-stimulated cells had little to no response to oligomycin where unstimulated, and both iNOS inhibitor treated BMDCs had significantly higher ATP-linked OCR values (Fig 5C). Thus, confirming that mitochondrial ATP production is restored when iNOS is inhibited in LPS-stimulated cells.

Figure 4. In this figure authors compared ECAR vs OCR with LPS and in the presence or absence of SEITU. They observed metabolic reprogramming in BMDC is NO-independent while sustained reprogramming is NO-dependent.

1. Fig 4B is not informative since Fig 4A itself explains no change in a glycolytic burst between the two groups (this can go to the supplementary figure). While Fig 2C is quantitatively good for understanding the exact difference between the two groups. This comparison is contained in the new Figure 6. We believe that showing the lack of a difference between the two groups with respect to acute glycolytic reprogramming is important context for the readers and have elected to keep these data in the main figure. We hope this acceptable for Reviewer#1.

2. Of curiosity did authors go longer than 600 minutes for ECAR and OCR and find any difference? We have previously run the assay to a full 12 hours and observed that at around 11 hours, the cells begin to exhibit metabolic fluctuation in OCR levels, which we take to be an indication of cellular instability at this point for extended incubation in a non-humidified chamber. This is the rationale for our limiting our data to shorter time periods displayed in our figure in this manuscript.

Reviewer #2: In the manuscript “Detection of nitric oxide-mediated metabolic effects using real-time extracellular flux Analysis” the authors established the phenomenon of an NO-dependent mitochondrial respiration threshold, which is linked to suppression of mitochondrial respiration in an NO-dependent manner. Authors further explored the efficacy of two different iNOS inhibitors in blocking the iNOS reaction kinetically in real time and explored parameters for application using Real Time Extracellular Flux Analysis. The experiments are conducted well but authors need to clarify a few points to strengthen the outcomes of the paper.

1) Is nitrite quantification an established method? Please provide few citations in the method section. We have now included multiple references to this point, found on lines 123-125 of the manuscript: “Quantification of NO indirectly via measurement of the stable intermediate nitrite is a validated and established method as first described here [28–33] as well as in macrophage and BMDC models here [14–16,34].”

2) Glycolytic burst is measured only based on pH. Did author consider measuring the concentration of extracellular and intracellular lactate? This will provide direct glycolytic flux because of the production of lactate. Kindly measure the lactate and Alanine concentration for the groups. Intracellular and extracellular lactate levels in LPS-stimulated BMDCs and macrophages have been extensively published and feel outside the domain of this work. However, we include mass spec data on these analytes for the reviewers consideration and confirmation below:

3) Make a schematic diagram of the overall research plan for this study. That will clearly demonstrate the various experimental events. The recommended schematic is now included as Figure 1 of the revised submission and described in lines 79-83 of the text: “Advances in immunometabolism technologies including the application of Real Time Extracellular Flux Analysis, colloquially referred to as “Seahorse” technology, have increased the specificity and flexibility to interrogate metabolic pathways involved in BMDC activation (Fig 1A-B) [4,5] as well as exploring NO-mediated effects on mitochondrial respiration (Fig 1C) [14–16].”

4) Did author measure the viability of cell at the end of OCAR and ECAR assays? Since these assays are really long, there are some chances that cells might chew up all of the essential components of media and become nutrition-deficient and might start slowing down metabolic activities and dying.

The 24-hour live/dead data from Figure 5F indicates that the cells are viable and healthy at these time points. In addition, the robust OCR levels (150 pmol/min) in the LPS stimulated group during these early time points indicate full respiratory function in these cells.

---

## [Decision Letter · Decision Letter 1]

8 Feb 2024

Detection of nitric oxide-mediated metabolic effects using real-time extracellular flux analysis

PONE-D-23-29008R1

Dear Dr. Amiel,

We’re pleased to inform you that your manuscript has been judged scientifically suitable for publication and will be formally accepted for publication once it meets all outstanding technical requirements.

Kind regards,

Nisha Singh, Ph.D.

Academic Editor

PLOS ONE

Additional Editor Comments (optional):

Reviewers' comments:

Reviewer's Responses to Questions

**Comments to the Author**

1. If the authors have adequately addressed your comments raised in a previous round of review and you feel that this manuscript is now acceptable for publication, you may indicate that here to bypass the “Comments to the Author” section, enter your conflict of interest statement in the “Confidential to Editor” section, and submit your "Accept" recommendation.

Reviewer #1: All comments have been addressed

Reviewer #2: All comments have been addressed

2. Is the manuscript technically sound, and do the data support the conclusions?

Reviewer #1: Yes

Reviewer #2: Yes

3. Has the statistical analysis been performed appropriately and rigorously? 

Reviewer #1: Yes

Reviewer #2: Yes

4. Have the authors made all data underlying the findings in their manuscript fully available?

Reviewer #1: Yes

Reviewer #2: Yes

5. Is the manuscript presented in an intelligible fashion and written in standard English?

Reviewer #1: Yes

Reviewer #2: Yes

6. Review Comments to the Author

Reviewer #1: My concerns have been adequately addressed and now the article connects all the gaps for the readers of PLOS ONE

Reviewer #2: The authors have responded to the comments properly. I recommend this manuscript to be published in this journal.

7. PLOS authors have the option to publish the peer review history of their article (what does this mean?). If published, this will include your full peer review and any attached files.

Reviewer #1: **Yes: **Manish Kumar

Reviewer #2: **Yes: **Dr. Rohit Mahar

---

## [Editor Report · Acceptance letter]

27 Feb 2024

PONE-D-23-29008R1 

PLOS ONE

Dear Dr. Amiel, 

I'm pleased to inform you that your manuscript has been deemed suitable for publication in PLOS ONE. Congratulations! Your manuscript is now being handed over to our production team.

Kind regards, 

on behalf of

Dr. Nisha Singh 

Academic Editor

PLOS ONE